# Advanced Glycation End-Product-Modified Heat Shock Protein 90 May Be Associated with Urinary Stones

**DOI:** 10.3390/diseases13010007

**Published:** 2025-01-02

**Authors:** Takanobu Takata, Shinya Inoue, Kenshiro Kunii, Togen Masauji, Junji Moriya, Yoshiharu Motoo, Katsuhito Miyazawa

**Affiliations:** 1Division of Molecular and Genetic Biology, Department of Life Science, Medical Research Institute, Kanazawa Medical University, Uchinada 920-0293, Ishikawa, Japan; 2Department of Pharmacy, Kanazawa Medical University Hospital, Uchinada 920-0293, Ishikawa, Japan; masauji@kanazawa-med.ac.jp; 3Department of Urology, Kanazawa Medical University, Uchinada 920-0293, Ishikawa, Japan or s-inoue@inoueiin-kusatsu.or.jp (S.I.); kenshiro@kanazawa-med.ac.jp (K.K.); 4Inoue Iin Clinic, Kusatsu 525-0034, Shiga, Japan; 5Department of General Internal Medicine, Kanazawa Medical University, Uchinada 920-0293, Ishikawa, Japan; moriya@kanazawa-med.ac.jp; 6General Medical Center, Kanazawa Medical University Hospital, Uchinada 920-0293, Ishikawa, Japan; 7Department of Internal Medicine, Fukui Saiseikai Hospital, Wadanaka 918-8503, Fukui, Japan; motoo.yoshiharu9082@fukui.saiseikai.or.jp

**Keywords:** ureteric stone, lifestyle-related disease, heat shock protein 90, advanced glycation end-products, MG-H1, argpyrimidine, GLAP, traditional Japanese medicine, urocalun, quercetin

## Abstract

Background: Urinary stones (urolithiasis) have been categorized as kidney stones (renal calculus), ureteric stones (ureteral calculus and ureterolith), bladder stones (bladder calculus), and urethral stones (urethral calculus); however, the mechanisms underlying their promotion and related injuries in glomerular and tubular cells remain unclear. Although lifestyle-related diseases (LSRDs) such as hyperglycemia, type 2 diabetic mellitus, non-alcoholic fatty liver disease/non-alcoholic steatohepatitis, and cardiovascular disease are risk factors for urolithiasis, the underlying mechanisms remain unclear. Recently, heat shock protein 90 (HSP90) on the membrane of HK-2 human proximal tubular epithelium cells has been associated with the adhesion of urinary stones and cytotoxicity. Further, HSP90 in human pancreatic and breast cells can be modified by various advanced glycation end-products (AGEs), thus affecting their function. Hypothesis 1: We hypothesized that HSP90s on/in human proximal tubular epithelium cells can be modified by various types of AGEs, and that they may affect their functions and it may be a key to reveal that LSRDs are associated with urolithiasis. Hypothesis 2: We considered the possibility that Japanese traditional medicines for urolithiasis may inhibit AGE generation. Of Choreito and Urocalun (the extract of *Quercus salicina* Blume/*Quercus stenophylla* Makino) used in the clinic, Choreito is a Kampo medicine, while Urocalun is a characteristic Japanese traditional medicine. As Urocalun contains quercetin, hesperidin, and *p*-hydroxy cinnamic acid, which can inhibit AGE generation, we hypothesized that Urocalun may inhibit the generation of AGE-modified HSP90s in human proximal tubular epithelium cells.

## 1. Introduction

Urinary stones (urolithiasis) have been categorized as kidney stones (renal calculus), ureteric stones (ureteral calculus and ureterolith), bladder stones (bladder calculus), and urethral stones (urethral calculus) [1,2,3,4]. Stones with calcium oxalate monohydrate and calcium phosphate crystals as the main components are generated in each region of the kidney, and move and adhere to glomerular and tubular cell membranes [3,5,6]. Although these stones induce cytotoxicity and inflammation in glomerular and tubular cells [1,2,3], along with strong pain [4,5,6], the mechanisms underlying their generation remain unclear. Some studies indicate that urinary stones are associated with lifestyle-related diseases (LSRDs) such as hyperglycemia [7], diabetes [8], non-alcoholic fatty liver disease (NAFLD)/non-alcoholic steatohepatitis (NASH) [9], and cardiovascular disease (CVD) [10]. However, the mechanisms involved in (i) urinary stone generation or promotion by LSRDs and (ii) the increased cytotoxicity from urinary stones in renal cells, because of LSRDs remain to be revealed. Yoodee et al. and Heng et al. recently indicated the possibility that (i) the growth of ureteric stones may be affected by heat shock protein 90 (HSP90) on the cell membrane, and that (ii) the adhesion of ureteric stones may promote HSP90 expression and induce cytotoxicity in the proximal tubular epithelium cells [11,12]. HSP90 contains two isomers, HSP90α and HSP90β; it maintains cellular homeostasis by acting as a molecular chaperone as well as by regulating autophagy and apoptosis [13,14,15,16]. HSP90 may, thus, be a key to understanding the relationship between urinary stones and LSRDs. Notably, advanced glycation end-products (AGEs) are associated with LSRDs, and some types of AGEs promote the symptoms and stages of LSRDs [17,18,19,20]. AGEs are generated from saccharides and their metabolic intermediate/sub-products (e.g., glyceraldehyde and methylglyoxal) react with proteins [17,18,19,20]. HSP90s in some cells which were treated with glyceraldehyde and methylglyoxal can be modified by some type of AGE structures (e.g., argpyrimidine, glyceraldehyde-derived pyridinium (GLAP), *N*^δ^-(5-hydro-5-methyl-4-imidazolone-2-yl)-ornithine (MG-H1), and an unidentified type of glyceraldehyde (GA)-derived AGE (GA-AGE)) [21,22,23]. We hypothesized that HSP90s on/in human proximal tubular epithelium cells can be modified by various types of AGEs, and that they may affect their functions and it may be the key to revealing that LSRDs are associated with urolithiasis. To inhibit AGE accumulation in cells, (i) AGE generation is inhibited via a carbonyl trap reaction system and glyoxalase-1 (GLO-1) activation [19,24,25], and (ii) AGE degradation (i.e., through autophagy and proteasomal degradation [26,27]) is promoted. In Japan, the following agents have been used to remove urinary stones or inhibit their generation: tamulosine [28], magnesium oxide [29], potassium citrate and sodium citrate hydrate [30], cystine-binding thiol drugs (e.g., tiopronin) [31], thiazide diuretics [32], febuxostat [33], Choreito (Chinese name: Zhu-Ling-Tang; Polyporus Decoction) [34,35], and Urocalun (The extract of *Quercus salicina* Blume/*Quercus stenophylla* Makino) [36,37]. Choreito is a Japanese traditional medicine (Kampo medicine) based on traditional Chinese medicine, while Urocalun is a characteristic Japanese traditional medicine, whose information has been inherited for some centuries in Japan [34,35,36,37]. We researched the components of these modern and traditional medicines and their ability to inhibit AGE generation or promote AGE degradation. Quercetin, hesperidine, and *p*-hydroxycinnamic acid in the extract of *Quercus salicina* Blume/*Quercus stenophylla* Makino are known to inhibit AGE generation via a carbonyl trap reaction system [19,25,38,39]. Thus, we hypothesized that Urocalun may inhibit the generation of AGE-modified HSP90s in human proximal tubular epithelium cells.

## 2. Urinary Stones

### 2.1. Types of Urinary Stones

Urinary stones (urolithiasis) have been categorized as kidney stones (renal calculus), ureteric stones (ureteral calculus and ureterolith), bladder stones (bladder calculus), and urethral stones (urethral calculus) (Figure 1) [1,2,3,4]. Renal and ureteral calculus are found in the upper urinary tract, while bladder and urethral calculus are found in the lower urinary tract. The main components of these stones are calcium oxalate monohydrate and calcium phosphate crystals, while struvite stones are composed of magnesium, ammonium, and phosphate [3]; these stones are generated in the respective regions of the kidney, and they move and adhere to the glomerular and tubular cell membrane [3,5,6]. Although four groups of stones have been categorized, they can be aggregated and combined through movement in the glomeruli and tubules [1,2,3,4,5,6]. Therefore, identifying the original location at which the stones were first generated is difficult, and their removal in clinical treatment is more common than this identification. Further, although these stones can adhere to the glomerular and tubular epithelial cells, the mechanism underlying their adhesion remains unclear [1,2,3,4,12]. We have previously described ureteric stones located in the proximal convoluted tubule [1,2,3,4]. Further, Heng et al. have reported that annexin A1, hyaluronic acid synthase 3 (HAS3), osteopontin, cluster of differentiation 44 (CD44), and HSP90 on HK-2 cells (human renal proximal convoluted tubule cell line) may act as receptors for nano-calcium oxalate monohydrate (nano-COM) in vitro (HSP90 is described in detail in Section 3) [12].

### 2.2. Urinary Stones (Urolithiasis) and LSRDs

LSRDs are a serious problem globally, especially in developed countries because of the consumption of excess nutrients, including saccharides, lipids, and proteins, as well as the lack of heavy manual labor [17,18,19]. Excess nutrient intake and calorie consumption results in obesity [40,41], promoting various LSRDs such as hyperglycemia and type 2 diabetes (T2DM) [42,43,44], NAFLD/NASH [44,45], CVD [16], cancer [46], and obstructive sleep apnea syndrome [47]. Some reports indicate that hyperglycemia [7,48,49], T2DM [8,50,51,52], NAFLD/NASH [9,53,54], and CVD [10,55,56,57] are risk factors for urinary stones (Figure 2). However, the mechanisms by which LSRDs promote urinary stone generation and their growth, their adhesion to cell membranes, and their cytotoxicity remain unclear. Furthermore, whether urinary stone progression promotes LSRDs also remains unexamined (Figure 2). Thus, various phenomena in LSRDs may affect urinary stones and these need to be examined to reveal the underlying relationships.

## 3. Urinary Stones and HSP90

### 3.1. Various Functions of HSP90 in Certain Cells

HSP90 exists in two isoforms, HSP90α (inducible form) and HSP90β (constitutively expressed form), and their amino acid sequences show approximately 85% similarity [14,15,16,58,59]. The heat shock-induced enhancement of HSP90α expression is more profound than that of HSP90β [14]; HSP90 is generated as a dimer and localized predominantly in the cytoplasm [15]. It is reported to exert various functions, including acting as a molecular chaperone [15,58,59,60,61,62] and regulating the ubiquitin–proteasome degradation system [63,64], autophagy [65,66], and apoptosis [23]. HSP90 dimers are needed to repair various proteins as a part of the molecular chaperone system. Although homodimers (HSP90α-HSP90α and HSP90β-HSP90β) are produced more commonly, heterodimers (HSP90α-HSP90β) can also be generated, and these HSP90 heterodimers are activated with adenosine triphosphate [58,59,60,61]. In this step, heat shock protein organizer protein (HOP) and heat shock protein 70 (HSP70) act as co-chaperones to generate the HSP90-HSP70-HOP protein complex [62]. This chaperone system can then regulate the ubiquitin–proteasome degradation system [62,63]. Proteins whose conformations are changed by the HSP90 dimer are degraded by this system in the second step [62,63]. In the autophagy system, activated HSP90 induces Atg7 and Beclin-1 expression to induce and upregulate microtubule-associated protein 1 light chain 3-II (LC3-II) [65,66]. Ectopic HSP90 on the plasma membrane and secreted HSP90 have also been reported even though HSP90 has been previously considered an intracellular protein in the past [67,68].

### 3.2. Adhesion of COM to Renal Proximal Tubule Epithelial Cells and the Expression and Function of HSP90

HSP90 is expressed in human renal proximal convoluted tubule epithelial cells [12,69]. Zhu et al. revealed that HSP90 expression in HK-2 human renal proximal tubule epithelial cells was less than that in 786-O and ACHN cells (other cell lines of human renal origin), and that AUY922, an inhibitor of HSP90, inhibited the proliferation of 786-O and ACHN cells [69]. HSP90 may also be a target of anti-tumor treatment. In contrast, Heng et al. suggested that (i) HSP90, CD44, annexin A1, HAS3, and osteopontin on the membrane of HK-2 cells may act as a receptor for nano-COM and that (ii) the adhesion of nano-COM with HK-2 membrane proteins induces the upregulation of HSP90 and CD44 (Figure 3) [12]. Although HSP90 is generally an intracellular protein [14,15,16,58,59], it can penetrate the membrane to act as a receptor, as reported in HK-2 cells [12]. The adhesion of nano-COM to HK-2 cells induces (i) an increase in reactive oxygen species (ROS), (ii) a decrease in G1-phase cells and an increase in S-phase cells, and (iii) the upregulation of phosphoserine (Figure 3) [12]. Owing to these phenomena, cell viability decreases. Although the expression of HSP90, CD44, annexin A1, HAS3, and osteopontin is increased under these conditions, the underlying mechanisms remain unknown [12]. HSP90 upregulation may, thus, be associated with cytotoxicity. Because the ratio of the HSP90 located on the membrane is increased, it remains unclear whether stimulation by nano-COM leads to increased HSP90 expression as a receptor. However, we believe that the increased HSP90 may induce various cellular maintenance and/or cytotoxicity processes [14,15,16,58,59].

## 4. Hypothesis for the Generation of AGE-Modified HSP90s in Human Proximal Tubular Epithelium Cells

### 4.1. Generation and Accumulation of AGEs

AGEs originate from saccharides (e.g., glucose and fructose) and their metabolic intermediates/sub-products (e.g., glyceraldehyde, glycolaldehyde, methylglyoxal, and glyoxal) (Figure 4) [17,18,19,20]. As glyceraldehyde is a triose, it is categorized as a saccharide even though it is produced through metabolism and a non-enzymatic reaction based on glucose and fructose [19,70,71]. In our previous study, Wistar/ST rats subjected to an intake of high-fructose corn syrup (fructose–glucose = 55:45) (HFCS) for 8 or 13 weeks showed the significant generation and accumulation of glyceraldehyde-derived AGEs (GA-AGEs) in their liver and blood (intracellular GA-AGEs were quantified using a slot blot analysis with Takata’s lysis buffer and a poly vinylidene difluoride (PVDF) membrane) [19,70,71,72]. We considered that most AGEs are generated in cells because glyceraldehyde, glycolaldehyde, methylglyoxal, glyoxal, and 3-deoxyglucosone are generally produced in cells and their reactivity is high. However, as these intermediates have also been detected and quantified in blood [73,74,75,76], AGEs can be generated in blood, and the category of extracellular AGEs needs to be corrected as (i) those secreted and leaked from cells, and (ii) those introduced from the intake of dietary AGEs [17,18,19]. Thus, the molecules shown in Figure 4 are classical AGEs based on their origin, while lactaldehyde and melibiose have been reported to give rise to novel AGEs (lactaldehyde-derived AGEs and melibiose-derived AGEs (MAGE)) [77,78].

We focused on glyceraldehyde and methylglyoxal because they show high reactivity and generate various AGE structures (Figure 5). Although AGEs are generated from saccharides and their metabolic intermediates/non-enzymatic reaction sub-products and proteins, the structures produced from the former are also named “AGEs” (free-type AGEs) (Figure 5) [19,79,80]. We introduce free-type AGEs containing one and two amino acid residues in Figure 5. *N*^ε^-carboxymethyl-lysine (CML), *N*^ε^-carboxyethyl-lysine (CEL), *N*^ε^-carboxyethyl-lysine, *N*^δ^-(5-hydroxy-5-methyl-4-imidazolone-2-yl)-ornithine (methylglyoxal-derived hydroimidazolone) (MG-H1), argpyrimidine, 3-hydroxy-5-hydroxymethyl-1-pyridinium (GLAP), and 6-{1-(5S)-5-ammonio-6-oxido-6-oxyohexyl-}-4-methyl-imidazolium-3-yl-L-norlecucine (methylglyoxal dimer) (MOLD) are generated from methylglyoxal [17,18,19,79,80]. In contrast, MG-H1, argpyrimidine, GLAP, trihydroxy-triosidine, and pyrrolopyridinium-lysine dimer 1 and 2 (PPG1 and 2) are derived from glyceraldehyde [17,18,19,79,80,81,82]. MG-H1, argpyrimidine, and GLAP have been revealed as AGEs capable of being generated from glyceraldehyde and methylglyoxal and thus belong to both GA-AGEs and methylglyoxal-derived AGEs (MGO-AGEs) (Figure 5) [17,19,79]. We previously reported the quantification of some types of GA-AGEs named as “Toxic AGEs (TAGEs)” by Takeuchi et al. using a slot blot analysis with Takata’s lysis buffer (or modified Takata’s lysis buffer) and a PVDF membrane [17,23,70,71,72]. The TAGEs named by Takeuchi et al. did not include major GA-AGEs such as MG-H1, argpyrimidine, GLAP, triosidines, PPG1, and PPG2. Although Takeuchi et al. hypothesized the two free-type AGE structures containing two and three amino acid residues, respectively, as the structures of TAGEs in 2024, this remains to be proved [83]. (Notes: Lee et al. demonstrated that extracellular MOLD combined with the receptor for AGEs (RAGE) and induced ROS production and mitochondrial dysfunction, and they, thus, named MOLD as a TAGE [84].) Furthermore, Shen et al. categorized GA-AGEs, MGO-AGEs, glycolaldehyde-derived AGEs (GO-AGEs), and 3-deoxyglucosone-derived AGEs (3-DG-AGEs) as TAGEs [85]. TAGEs, as described by Shen, belong to a wide range of various AGEs.

### 4.2. The Model of Identified and Predicted AGE-Modified HSP90s

We considered the crude AGE pattern in this study (as certain metabolites or non-enzymic reaction products can generate various AGE structures) (Figure 6). These patterns were applied to AGE-modified HSP90 (Figure 6) [19,79,80]. Norkin et al. reported that argpyrimidine and MG-H1-modified HSP90 were generated in a human breast cancer cell line (MDA-MB-231 cells) (Figure 6 and Figure 7) [21]. They performed a Western blot analysis using anti-argpyrimidine and anti-MG-H1 antibodies for the immunoprecipitation samples of HSP90 in MDA-MB-231 cells treated with methylglyoxal. Furthermore, they analyzed their samples using high-performance liquid chromatography (HPLC)–electrospray ionization (ESI)–mass spectrometry (MS) (HPLC-ESI-MS), and detected some peptides which were CEL-, argpyrimidine-, and hydroimidazolone-modified. (We considered that this hydroimidazolone was MG-H1 (Appendix A) [21].) In contrast, Senavirathna et al. reported that MG-H1, argpyrimidine, and GLAP were generated in a human pancreatic ductal cell line (PANC-1) treated with glyceraldehyde [22]. Using HPLC-ESI-MS, they also determined that HSP90β in PANC-1 cells was modified with MG-H1, argpyrimidine, and GLAP. The data obtained by Norkin et al. and Senavirathna et al. proved that various types of AGE-modified HSP90s could be generated (Figure 6).

We previously reported that some types of GA-AGEs, which Takeuchi et al. named as TAGEs, were generated in PANC-1 cells treated with glyceraldehyde [23]. These GA-AGEs were quantified using a slot blot analysis with Takata’s lysis buffer and PVDF membrane and showed glyceraldehyde dose-dependent accumulation. Notably, high molecular weight HSP90β (HMW-HSP90β) was generated and increased with glyceraldehyde dose-dependence as determined using a Western blot analysis [23]. Although HMW-HSP90β may be a type of GA-AGE with a multiple AGE pattern wherein the glyceraldehyde-derived material combined with HSP90β and a certain protein (HSP90β or another protein), we were unable to identify this structure. Although GA-AGEs that combine two amino acid residues have been reported (Figure 5), we cannot prove that the individual GA-AGE structure combines two proteins based on the Western blot analysis with their antibodies alone because antibodies cannot recognize intermolecular or intramolecular covalent bonding (Figure 7). If we reveal that an antibody against one type of GA-AGE recognizes and binds to HSP90 using the Western blot analysis, these data will be insufficient to prove that the structure of GA-AGEs combines two proteins bound via an intermolecular covalent bond (Figure 7). Although AGE-modified HSP90s are detected in human pancreatic ductal and mammary gland cells, they have not been detected in human kidney cells such as glomerular and tubular cells.

### 4.3. AGEs and LSRDs

Various AGEs have been associated with LSRDs such as hyperglycemia, T2DM, NAFLD/NASH, and CVD [17,18,19,70,71,79,80]. Kehm et al. reported that argpyrimidine and pentosidine were accumulated in the pancreatic tissue of New Zealand Obese/H1BomDife mice fed a carbohydrate-rich diet [86]. Morioka et al. revealed that GA-AGEs and MGO-AGEs were accumulated in the pancreatic islets, in the α and β cells of STZ-induced diabetic Wistar rats, respectively [87]. Zhang et al. reported the presence of both AGEs in the blood and oil in the hepatic tissues of KK-Ay mice used to model T2DM with NASH [88]. In the cardiac tissues of older people, ryanodine receptor 2 (RyR2) is modified with 2-ammnonio-6-[4-(hydroxymetyl)-3-oxidopyridinium-1-yl]-hexanoate-lysine (4-hydroxymethyl-OP-lysine, hydroxymethyl-OP-lysine) and various other MGO-AGEs, and may induce the excessive leakage of Ca^2+^ from glycated RyR2 [19]. In contrast, F-actin–tropomyosin filaments are modified with glyoxal-derived hydroimidazolone (GH-1) and MG-H1, and may induce a reduction in contractile force [19]. In contrast, Wang et al. reported that the cardiomyocytes in Sprague Dawley (SD) rats subjected to an intake of GA-AGE-modified bovine serum albumin (GA-AGEs-BSA) underwent apoptosis, and that cultured cardiomyocytes cells treated with GA-AGEs-BSA also underwent apoptosis [89]. Thus, GA-AGEs-BSA may act as dietary AGEs and AGEs in body fluids such as blood.

### 4.4. Hypothesis for AGE-Modified HSP90s in Human Proximal Tubular Epithelium Cells and Their Functions

Various AGEs-modified HSP90s have been identified in some cultured cells treated with glyceraldehyde and methylglyoxal which increased in the various organs and cells in the patients with LSRDs [17,18,19,20]. We categorized these into type 1 diverse AGE patterns (certain AGE structures that can modify the same or different amino acids in one type of protein, and type 1 multiple AGE patterns (certain AGEs that modify a single protein molecular but not a specific type of protein (one protein molecule but not one type of protein) (Appendix A) [19,79,80]. Norkin et al. reported various modifications of the AGE structure in human recombinant HSP90 and naturally occurring HSP90 in MDA-MB-231 cells treated with methylglyoxal, which belong to type 1 diverse and type 1 multiple AGE patterns. We introduce some of the peptides analyzed by Norkin et al. using HPLC-ESI-MS in Appendix A. They suggested that various modification spots in an amino acid sequence act as molecular chaperones and as reporters for adenosine triphosphate (ATP) [21].

Various AGE modifications may induce dysfunction or excess function resulting in cytotoxicity. We considered the possibility of the amino acids in HSP90 on the plasma membrane being affected. Glyceraldehyde, glycolaldehyde, methylglyoxal, and glyoxal can move through body fluids like blood [73,74,75,76]. They may also be able to move through urine and generate various GA- and MGO-AGE-modified HSP90 molecules on the plasma membrane of renal cells. Further, intracellular AGE-modified HSP90 may be able to transfer to the plasma membrane. Because Yoodee et al. suggested that HSP90s were associated with the generation and growth of kidney stones [11], and we hypothesized that the AGE modification of HSP90s on the cellular membranes of the renal proximal tubular epithelium cells may affect their functions and the generation/growth of kidney stones. In contrast, the adhesion of COM induces the upregulation of HSP90 in HK-2 cells [12]. Although intracellular HSP90 is generated via stimulation from COM adhesion, it is modified by various types of AGEs. In this environment, HSP90 expression may be upregulated because a normal type of HSP90 is needed to maintain cell survival. Therefore, we hypothesized that the generation and accumulation of AGE-modified HSP90s in the renal proximal tubular epithelium cells may affect their functions. If AGE-modified HSP90s affect cellular homeostasis against the stimulation of the adhesion of kidney stones, it may be a key to reveal that LSRDs are associated with urolithiasis.

### 4.5. Inhibition of the Generation/Accumulation of AGEs Contained in AGE-Modified HSP90

To avoid the accumulation of AGEs in cells, we can select major methods such as (i) the inhibition of AGE generation [19,25] and (ii) the degradation of AGEs [26,27]. The former further includes two methods: (a) the carbonyl trap system [90,91,92] and (b) activation of glyoxalase-1 (GLO-1) [93,94,95]. Because GLO-1 metabolizes glyoxal and methylglyoxal, its activation can inhibit the generation of GO- and MGO-AGEs. Although aminoguanidine can inhibit the generation of various AGEs via the carbonyl trap system, it is unsuitable for medical use because of its high cytotoxicity [90,91]. Several studies have attempted to identify natural compounds that inhibit the generation of intracellular AGEs in medicinal plants because they will be suitable and safe for human use [92,93,96,97]. Quercetin [19,25,98], hesperidin [19,99], *p*-hydroxy cinnamic acid [19,25,100], resveratrol [19,25,101,102,103], and curcumin [19,103] have been reported to inhibit the generation of AGEs (Figure 8). All these natural compounds have the function of the carbonyl trap system, and both resveratrol and curcumin can activate GLO-1. The modification of AGEs derived from glyceraldehyde, glycolaldehyde, methylglyoxal, and glyoxal against HSP90 may, thus, be inhibited by the carbonyl trap system and GLO-1 activation.

## 5. Hypothesis of Medicines for Urolithiasis Against the Generation/Accumulation of AGE-Modified HSP90

### 5.1. Preventive and Therapeutic Medication for Urolithiasis

To treat or prevent urolithiasis, some modern and traditional medicines have been selected in Japan. Tamulosine [28,104], magnesium oxide [29,105], potassium citrate and sodium citrate hydrate [30,106], cystine-binding thiol drugs (e.g., tiopronin) [31,107], thiazide diuretics [32,108], and febuxostat [33,109] belong to the category of modern medicines. These act by (i) blocking the α1 receptor and inducing expansion of the proximal convoluted tubule [28,104]; (ii) the production of magnesium oxalate monohydrate but not urinary stones (calcium oxalate monohydrate and calcium phosphate crystals) [29,105]; (iii) the promotion of alkaluria [30,106] and urinary stone dissolution in urine [31,107]; and (iv) the inhibition of Ca^2+^ transport into urine [32,108] and inhibition of xanthinoxidase to suppress uric acid production [33]. In addition, several researchers, doctors, and pharmacists have focused on traditional medicines used worldwide because modern medicines (e.g., low-molecular organic compounds and antibodies) show several limitations against various diseases, whereas the use of traditional medicines is based on the clinical data from some hundred to thousand years [92,93,110,111,112]. In Japan, Choreito (Chinese name: Zhu-Ling-Tang; Polyporus Decoction) [34,35] and Urocalun (the extract of *Quercus salicina* Blume/*Quercus stenophylla* Makino) [36,37] have been selected as the traditional medicines for urolithiasis. Choreito, a Kampo medicine (classical Japanese traditional medicine) whose scientific/clinical data are based on Chinese traditional medicine, has been modified in Japan [34,35]. Researchers can access the database named Standard of Reporting Kampo Products (STORK) to research the crude drugs contained in Kampo medicines [113,114]. The extract of *Quercus salicina* Blume/*Quercus stenophylla* Makino is not included in Kampo medicines, and is a characteristic Japanese traditional medicine because people in the western area of Japan have used it to treat urolithiasis for some hundred years [36,37,38,39]. The mechanisms of both Choreito and Urocalun may promote urinary stone removal from the renal tubule by increasing the urinary volume.

### 5.2. Hypothesis for Urocalun Treatment for Inhibiting the Generation/Accumulation of AGE-Modified HSP90

Although we examined the components of various modern and traditional medicines in Japan, we could not find desirable components that could inhibit the generation of intracellular AGEs except for Urocalun. We, thus, believe that Urocalun may be able to inhibit the intracellular AGEs contained in AGE-modified HSP90 because the extract of *Quercus salicina* Blume/*Quercus stenophylla* Makino contains quercetin, hesperidin, and *p*-hydroxycinnamic acid (Figure 8) [36,37,38,39]. These three compounds can inhibit the generation of intracellular AGEs. They may be metabolized through hydroxylation and glycosylation before they are transported in renal proximal tubule epithelium cells, and it remains unclear [18]. However, we hypothesized that quercetin, hesperidin, and *p*-hydroxycinnamic acid may inhibit the AGE modification of HSP90 under the condition that they are transported into renal cells because carbonyl trap systems can work against compounds with aldehyde and ketone groups (origins of free-type AGEs such as glyceraldehyde, glycolaldehyde, methylglyoxal, and glyoxal), and suppress their reaction with universal proteins (Figure 4).

## 6. Limitations

### 6.1. Various Types of Compounds for the Promotion or Inhibition of Kidney Stone Growth

In this article, we introduced HSP90, HAS3, annexin A1, CD44, and osteopontin as the receptors for kidney stones based on the study by Heng et al. [12]. However, several other compounds have been reported as promoters or inhibitors for the growth of kidney stones for approximately 70 years [115,116,117,118,119,120,121]. Glycosaminoglycans (e.g., heparan sulfate, chondroitin sulfate, and hyaluronic acid) [115,116,117], sodium pentosan polysulfate [118], saponin [119], surfactant [120], and citrate [30,106] can inhibit the growth of kidney stones and pyrophosphate treatment reduces renal calcifications [121]. The functions of urinary proteoglycans are complex. Although glycosaminoglycans were promoters of COM crystal nucleation, they are inhibitors of COM aggregation [117]. In contrast, magnesium pyrophosphate kidney stones have been reported in clinical research, suggesting that pyrophosphates can generate kidney stones [122]. The relationships between these compounds and AGEs remain unclear. As proteoglycans, which are the components of cell membranes and the extracellular matrix, contain protein, they may undergo AGE modification. However, we have not obtained data indicating that proteoglycans are modified by free-type AGE structures such as CML, CEL, and MG-H1 (Figure 5). In contrast, hyaluronan is produced by HAS3, and binds CD44 to induce the generation of a “Hyaluronic coat” [123,124]. However, whether this hyaluronic coat can inhibit the AGE modification of HSP90 or suppress the cytotoxicity of AGE-modified HSP90 remains unclear.

### 6.2. CML in the Kidney and Intracellular/Extracellular Glucose-Derived AGEs in the Renal Proximal Tubule Epithelial Cells

Maejima et al. reported that autophagy controls lysosomes [125], and Takahashi et al. revealed that AGEs are degraded by lysosomes, which are controlled by autophagy systems based on previous investigations [126]. Takahashi et al. reported that (i) CML was significantly accumulated in the kidney of STZ-diabetic model mice with the insufficiency of autophagy systems, (ii) extracellular AGEs (origin, glucose) and high-glucose medium which induce the generation of intracellular AGEs promoted the biosynthesis of lysosomes in renal proximal tubule epithelial cells, (iii) these AGEs were degraded by lysosomes, and (iv) lysosomes were upregulated by autophagy systems [126]. Although these findings were reported, we cannot confirm the possibility of their association with AGE-modified HSP90.

## 7. Conclusions

Although the cytotoxicity mechanism of urinary stones remains unclear, HSP90 expression in human renal proximal tubule epithelium cells may be targeted for both the adhesion of urinary stones and the associated cellular homeostasis. In contrast, HSP90s are able to be modified by various types of AGEs such as MG-H1, argpyrimidine, and GLAP. We hypothesized that AGE-modified HSP90s may, thus, be the key to revealing the relationship between LSRDs and urolithiasis. Furthermore, we hypothesized that Urocalun, which contains quercetin, hesperidin, and *p*-hydroxy cinnamic acid, may inhibit the generation of intracellular AGE-modified HSP90s.

## Figures and Tables

**Figure 1 diseases-13-00007-f001:**
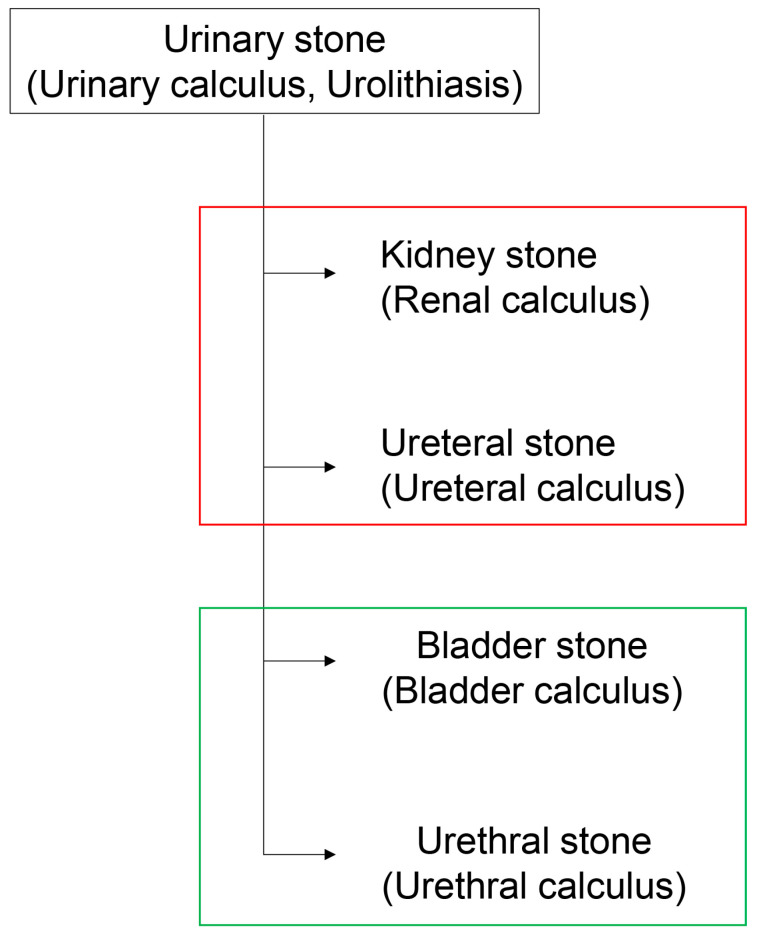
Urinary stones (urolithiasis) include kidney stones, ureteral stones, bladder stones, and urethral stones. The open red box indicates the upper urinary tract. An open green box indicates the lower urinary tract.

**Figure 2 diseases-13-00007-f002:**
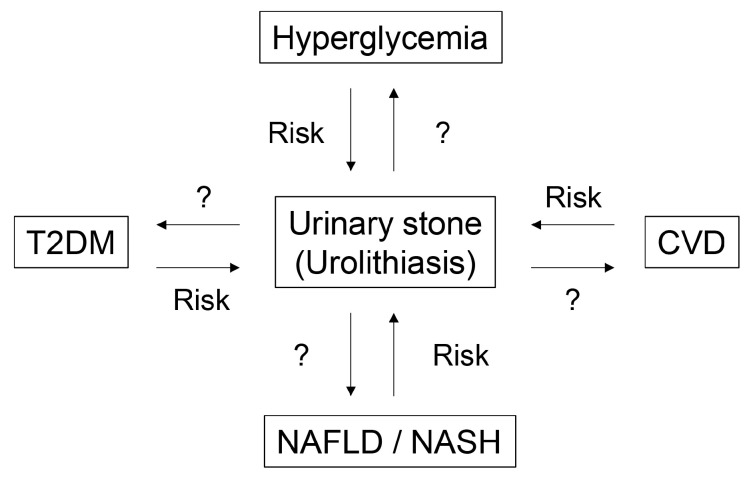
The relationship between urinary stones and lifestyle-related diseases. T2DM, type 2 diabetes mellites; CVD, cardiovascular disease; NAFLD, non-alcoholic fatty liver disease; NASH, non-alcoholic steatohepatitis.

**Figure 3 diseases-13-00007-f003:**
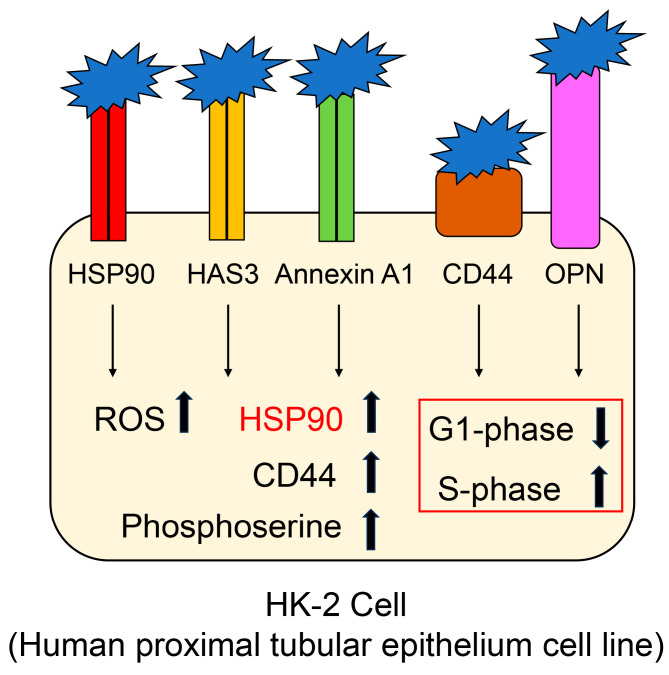
Nano-COM binds to five proteins acting as receptors and induces various processes in HK-2 cells. HSP90, HAS3, annexin A1, CD44, and osteopontin located on the membrane act as its receptors. Closed blue prickly leaves indicate nano-COM. The closed red, orange, green, brown, and pink squares indicate HSP90, HAS3, annexin A1, CD44, and OPN, respectively. The open red square shows the cell cycle effects. COM, calcium oxalate monohydrate; HSP90, heat shock protein 90; HAS3, hyaluronic acid synthase 3; CD44, cluster of differentiation 44; OPN, osteopontin. ROS, reactive oxygen species.

**Figure 4 diseases-13-00007-f004:**
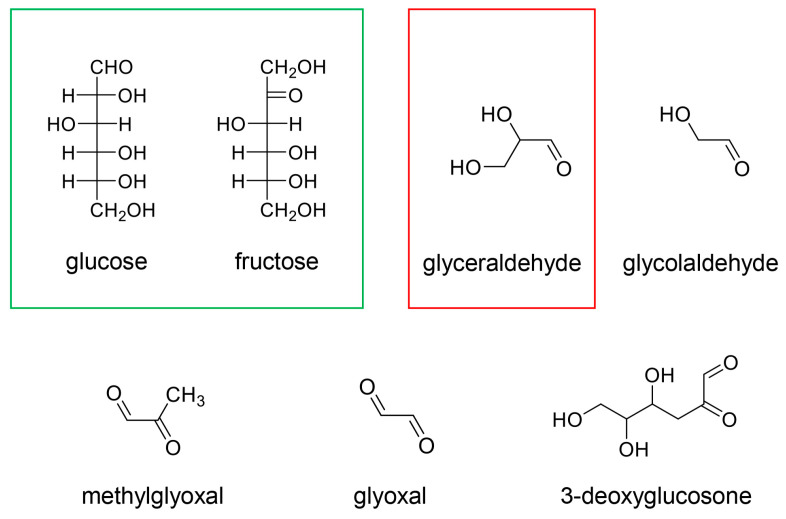
The origin of AGEs. The open green box indicates hexoses. The open red box indicates a triose. Glucose, fructose, and glyceraldehyde are saccharides. In contrast, glyceraldehyde, glycolaldehyde, methylglyoxal, glyoxal, and 3-deoxyglycosone are the metabolic intermediates/non-enzymatic reaction sub-products of glucose and fructose.

**Figure 5 diseases-13-00007-f005:**
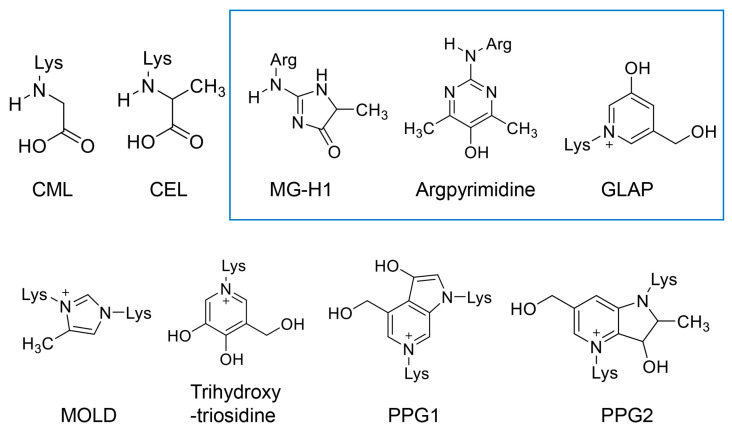
Free-type AGEs containing one or two amino acid residues. The AGEs in the upper tier contain one amino acid residue. The AGEs in the lower tier contain two amino acid residues. Lys, lysine; Arg, arginine [19,80]; CML, *N*^ε^-carboxymethyl-lysine [17,18,19,80]; *N*^ε^-carboxyethyl-lysine [17,18,19,80]; MG-H1, *N*^δ^-(5-hydrox-5-methyl-4-imidazolone-2-yl)-ornithine (methylglyoxal-derived hydroimidazolone) [17,18,19,79,80]; GLAP, 3-hydroxy-5-hydroxymethyl-1-pyridinium [17,19,79,80]; MOLD, 6-{1-(5S)-5-ammonio-6-oxido-6-oxyohexyl-}-4-methyl-imidazolium-3-yl-L-norleucine (methylglyoxal dimer) [17,19,80]. PPG1 and 2, pyrrolopyridinium-lysine dimer derived from glyceraldehyde 1 and 2 [19,82]. The open blue box indicates free-type AGEs generated from both glyceraldehyde and methylglyoxal [18,19,79].

**Figure 6 diseases-13-00007-f006:**
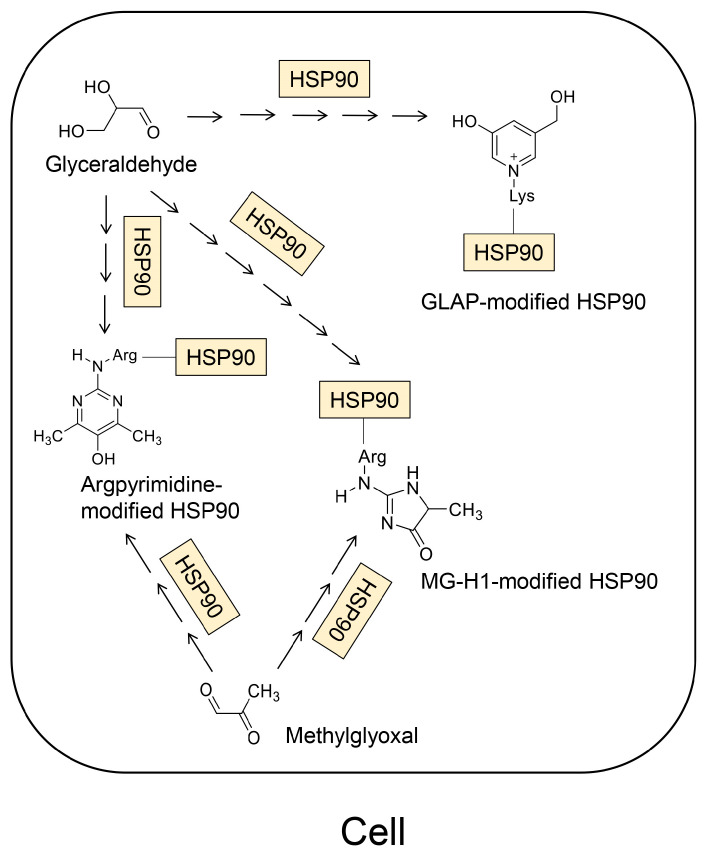
MG-H1-, argpyrimidine-, and GLAP-modified HSP90s can be generated from glyceraldehyde and/or methylglyoxal in the cell [21,22]. Lys, lysine; Arg, arginine; HSP90, heat shock protein 90. The arrows show a step of the Maillard and a non-enzymatic reaction between methylglyoxal and an amino acid residue (lysine or arginine) in HSP90.

**Figure 7 diseases-13-00007-f007:**
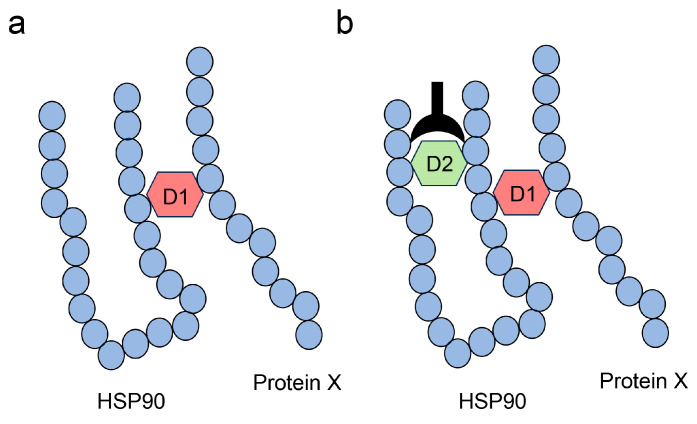
Model of the free-type AGE combined with HSP90 and another protein (protein X) via an intermolecular covalent bond, and with the amino acid in HSP90 via an intramolecular covalent bond. HSP90, heat shock protein 90. D1 and D2 indicate the AGEs capable of binding two amino acid residues simultaneously. (**a**) D1 combines HSP90 and protein X via an intermolecular covalent bond. (**b**) D1 combines two amino acids in both HSP90 and protein X via intermolecular covalent bonds, while D2 combines two amino acids in HSP90 via intramolecular covalent bonds. The closed black plow represents an anti-D2-antibody in HSP90.

**Figure 8 diseases-13-00007-f008:**
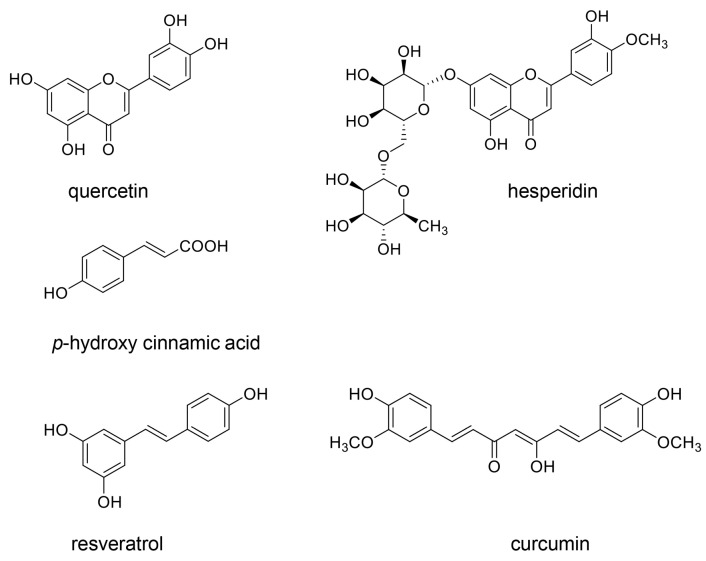
Natural compounds that inhibit the generation of intracellular AGEs. Quercetin [19,25,98], hesperidin [19,99], *p*-hydroxy cinnamic acid [19,25,100], resveratrol [19,25,101,102,103], and curcumin [19,103] have the function of the carbonyl trap system. Both resveratrol and curcumin can activate GLO-1.

## Data Availability

The data presented in this study are available from the corresponding author upon request.

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
