# Peer review of "Advanced Glycation End-Product-Modified Heat Shock Protein 90 May Be Associated with Urinary Stones"

_diseases, 2025, doi:10.3390/diseases13010007_

Round 1
Reviewer 1 Report
Comments and Suggestions for Authors
Main points:
1. Concerning urolithiasis promoters and inhibitors, hundreds of papers were published in important journals, such as Kidney Int., Nephron, and many others, from 1953 to 2024 on the role of glycosaminoglycans and other compounds, such as pyrophosphate, citrate, pentosan polysulfate, saponins, and surfactants, as inhibitors of crystallization. This relevant aspect should at least be mentioned in the present review. As proteoglycans (and their glycosaminoglycan chains) are important components of cell surface and extracellular matrix, their biological functions may be affected by AGEs. Did the authors consider this possibility?
2. Lines 86-87 – stone composition: struvite stones are also very important, and their crystallization process is different. This type of stones should also be considered here.
3. Lines 143-145 – It was already shown that the proximal tubular cells are the first affected by AGEs in diabetic nephropathy. Other mechanisms besides HSP90 may be involved, such as lysosomal enzymes. In other words, HSP90 may not be solely responsible for the observed results. This point should also be considered.
4. Line 155 – although the authors cited CD44 and HAS3, they did not consider hyaluronan (the glycosaminoglycan synthesized by HAS3 and the ligand of CD44). Is it possible that the increased expression of HAS3 and CD44 could induce an increased hyaluronan “coat” on HK-2 cells, protecting these cells from the AGEs actions?
Minor point:
Line 52 – “stones may affected” – should be “stones may be affected”.
Author Response
Response Letter to Reviewers’ Comments
Responses to Reviewer 1
Dear Reviewer 1:
Thank you for giving us the opportunity to submit a revised draft of our manuscript titled “Advanced Glycation End-Products-modified Heat Shock Protein 90 May be Associated with Urinary Stones” to the Diseases (manuscript ID: 3335984). We appreciate the time and effort you have taken to provide valuable feedback on our manuscript; your comments have enriched the manuscript and helped us produce a more balanced account of our research. The manuscript has been reviewed by a professional English editor (Editage) to address all grammatical and syntax errors and improve the overall readability of the document.
We have inserted a new Section 6 (Limitation), and removed the previous Reference 3 and inserted the new references 3 and 115-126.
Comments and Suggestions for Authors
Main points:
Comment 1: Concerning urolithiasis promoters and inhibitors, hundreds of papers were published in important journals, such as Kidney Int., Nephron, and many others, from 1953 to 2024 on the role of glycosaminoglycans and other compounds, such as pyrophosphate, citrate, pentosan polysulfate, saponins, and surfactants, as inhibitors of crystallization. This relevant aspect should at least be mentioned in the present review. As proteoglycans (and their glycosaminoglycan chains) are important components of cell surface and extracellular matrix, their biological functions may be affected by AGEs. Did the authors consider this possibility?
Response 1: Thank you for bringing these points to our attention. We have included the following sentences in newly added Section 6.1. (Limitation)
In this article, we introduced HSP90, HAS3, annexin A1, CD44, and osteopontin as the receptors for kidney stones based on the study by Heng et al. [12]. However, several other compounds have been reported as promoters or inhibitors for the growth of kidney stones since approximately 70 years [115–121]. Glycosaminoglycans [115–117], sodium pentosan polysulfate [118], saponin [119], surfactant [120], citrate [30,106] can inhibit the growth of kidney stones and pyrophosphate treatment reduces renal calcifications [121]. In contrast, magnesium pyrophosphate kidney stones have been reported in clinical research, suggesting that pyrophosphates can generate kidney stones [122]. The relationships between these compounds and AGEs remain unclear. As proteoglycans, which are the components of cell membranes and the extracellular matrix, contain protein, they may undergo AGE-modification. However, we have not obtained data indicating that proteoglycans are modified by free-type AGE structures such as CML, CEL, and MG-H1 (Figure 5).
Accordingly, we have inserted the new references 115-122.
Comment 2: Lines 86-87 – stone composition: struvite stones are also very important, and their crystallization process is different. This type of stones should also be considered here.
Response 2: We have inserted the sentence for struvite in Section 2.1., and inserted the new reference 3.
Comment 3: Lines 143-145 – It was already shown that the proximal tubular cells are the first affected by AGEs in diabetic nephropathy. Other mechanisms besides HSP90 may be involved, such as lysosomal enzymes. In other words, HSP90 may not be solely responsible for the observed results. This point should also be considered.
Response 3: We understand that Takahashi et al. reported that (i) CML were significantly accumulated in the kidneys of STZ-diabetic model mice with insufficiency of autophagy systems, (ii) extracellular AGEs (origin, glucose) and high-glucose medium which induce the generation of intracellular AGEs promoted the biosynthesis of lysosomes in a renal proximal tubule epithelial cell line, (iii) these AGEs were degraded by lysosomes and (iv) lysosomes were upregulated by autophagy systems. Therefore, we have introduced this information in Section 6.2. and inserted references 125 and 126.
Comment 4: Line 155 – although the authors cited CD44 and HAS3, they did not consider hyaluronan (the glycosaminoglycan synthesized by HAS3 and the ligand of CD44). Is it possible that the increased expression of HAS3 and CD44 could induce an increased hyaluronan “coat” on HK-2 cells, protecting these cells from the AGEs actions?
Response 4: We added the following sentences to Section 6.1.
In contrast, hyaluronan is produced by HAS3, and binds CD44 to induce generation of a “Hyaluronic coat” [123,124]. However, whether this hyaluronic coat can inhibit the AGE-modification of HSP90 or suppress the cytotoxicity of AGE-modified HSP90 remains unclear.
More, we have inserted the new references 123 and 124.
Minor point:
Comment 1: Line 52 – “stones may affected” – should be “stones may be affected”.
Response 1: We have corrected this sentence.
Reviewer 2 Report
Comments and Suggestions for Authors
In the present manuscript Authors examine the literature evidence that some live-style related diseases that promote the formation of AGE may favor the adhesion of AGE-modified HSP90 molecules to the plasma membrane of renal and bladder cells, thus inducing the formation of renal or urethra stones. Then, some natural compounds found in traditional Japanese medicine are suggested to prevent the formation of AGE-bound HSP90 molecules on the surface of renal and ureteric cells, thus preventing or even curing urinary stones.
The sections describing the relation of life styles with stone formation and the expression and function of HSP90 are rather well developed. Unfortunately, section 4.1, describing the formation of AGE, is written in a rather confused way; moreover, it appears to be truncated. In my opinion, section 4.1 might be very useful for a reader wishing to understand the chemistry of AGE formation, but it is not manageable unless it is re-written with the help of an English mother-language expert. Section 4.2 is a bit more understandable, but still needs a lot of refinement. The remaining sections are highly speculative in character, although they appear to be well substantiated.
In my opinion, this manuscript advances a very reasonable hypothesis, but it is a pity that the hypothesis is not accompanied by experimental data. Authors should try to show,at least, that (i) in the presence of oxidative stress conditions, AGE-linked HSP90 is found on the surface of cultured renal cells; and (ii) the treatment with any of the compounds of traditional medicine is able to prevent the formation of AGE or the binding of AGE to HSP90 or HSP90 localization on the plasma membrane of relevant cells.
Comments on the Quality of English LanguageAs already specified above, the quality of English greatly limits the understanding of section 4.1 and affects other sections as well
Author Response
Response Letter to Reviewers’ Comments
Responses to Reviewer 2
Dear Reviewer 2:
Thank you for giving us the opportunity to submit a revised draft of our manuscript titled “Advanced Glycation End-Products-modified Heat Shock Protein 90 May be Associated with Urinary Stones” to the Diseases (manuscript ID: 3335984). We appreciate the time and effort you have taken to provide valuable feedback on our manuscript; your comments have enriched the manuscript and helped us produce a more balanced account of our research. The manuscript has been reviewed by a professional English editor (Editage) to address all grammatical and syntax errors and improve the overall readability of the document.
We have inserted a new Section 6 (Limitation), and removed the previous Reference 3 and inserted the new references 3 and 115-126.
Comments and Suggestions for Authors
In the present manuscript Authors examine the literature evidence that some live-style related diseases that promote the formation of AGE may favor the adhesion of AGE-modified HSP90 molecules to the plasma membrane of renal and bladder cells, thus inducing the formation of renal or urethra stones. Then, some natural compounds found in traditional Japanese medicine are suggested to prevent the formation of AGE-bound HSP90 molecules on the surface of renal and ureteric cells, thus preventing or even curing urinary stones.
Comment 1: The sections describing the relation of life styles with stone formation and the expression and function of HSP90 are rather well developed. Unfortunately, section 4.1, describing the formation of AGE, is written in a rather confused way; moreover, it appears to be truncated. In my opinion, section 4.1 might be very useful for a reader wishing to understand the chemistry of AGE formation, but it is not manageable unless it is re-written with the help of an English mother-language expert. Section 4.2 is a bit more understandable, but still needs a lot of refinement. The remaining sections are highly speculative in character, although they appear to be well substantiated.
Response 1: We have corrected the sentences in Section 4.1. and 4.2., and an English Proofreader from Editage has checked and corrected the sentences in our manuscript.
Comment 2: In my opinion, this manuscript advances a very reasonable hypothesis, but it is a pity that the hypothesis is not accompanied by experimental data. Authors should try to show, at least, that (i) in the presence of oxidative stress conditions, AGE-linked HSP90 is found on the surface of cultured renal cells; and (ii) the treatment with any of the compounds of traditional medicine is able to prevent the formation of AGE or the binding of AGE to HSP90 or HSP90 localization on the plasma membrane of relevant cells.
Response 2: We have added the following sentences to Section. 4.2.
We cannot confirm that AGE-modified HSP90 are generated in human renal proximal tubule epithelial cells of patients with LSRDs. However, we believe that AGE-modification of HSP90 in both normal and LSRD conditions occurred because HSP27 in the human epithelial gastric cells (cultured cell line), which was incubated with normal glucose levels (5.5 mM), was modified with argpyrimidine (Figure 5) [18]. We believe that various proteins such as HSP90 can undergo AGE-modification under normal glucose conditions, and that this modification is promoted under high glucose and fructose levels and under oxidative stress condition because the source compounds of free type-AGEs such as methylglyoxal, glyceraldehyde, and glyoxal are increased (Figure 4) [19].
We described the following sentences to Section 5.2.
These three compounds can inhibit the generation of intracellular AGEs. They may be metabolized through hydroxylation and glycosylation before they are transported in renal proximal convoluted tubule epithelial cells, and it remain unclear [18]. However, if quercetin, hesperidin, and p-hydroxycinnamic acid are transported in renal cells, they may inhibit AGE-modification of HSP90 because carbonyl trap systems can work against compounds with aldehyde and ketone groups (origins of free-type of AGEs such as glyceraldehyde, glycolaldehyde, methylglyoxal and glyoxal), and suppress their reaction with universal proteins (Figure 4).
Comments on the Quality of English Language
As already specified above, the quality of English greatly limits the understanding of section 4.1 and affects other sections as well
Response: We have corrected the sentences in Section 4.1., and an English Proofreader from Editage has checked and corrected sentences in our manuscript.
Round 2
Reviewer 1 Report
Comments and Suggestions for Authors
Although most of our previous concerns were addressed, some aspects remain. For instance, the new reference on glycosaminoglycans (#115) is inadequate. The cited article is about corneal glycosaminoglycans, and as everyone knows, the cornea is a non-irrigated tissue, whose glycosaminoglycan composition contributes very little (or nothing) to urinary glycosaminoglycans. There are many articles reporting changes in urinary glycosaminoglycan excretion in urolithiasis. Please check again and replace this reference with one that is more appropriate to the subject. Citing this reference suggests that the authors are unfamiliar with "glycosaminoglycans".
Author Response
Response Letter to Reviewers’ Comments
Responses to Reviewer 1
Dear Reviewer 1:
Thank you for giving us the opportunity to submit a revised draft of our manuscript titled “Advanced Glycation End-Products-modified Heat Shock Protein 90 May be Associated with Urinary Stones” to the Diseases (manuscript ID: 3335984). We appreciate the time and effort you have taken to provide valuable feedback on our manuscript; your comments have enriched the manuscript and helped us produce a more balanced account of our research.
Because Editorial Office permitted the change of article type, our previous Review article has been changed to the “Hypothesis article”.
We removed and described some sentences in 2nd revised manuscript. Please read them with Yellow Highlighted.
Comment 1: Although most of our previous concerns were addressed, some aspects remain. For instance, the new reference on glycosaminoglycans (#115) is inadequate. The cited article is about corneal glycosaminoglycans, and as everyone knows, the cornea is a non-irrigated tissue, whose glycosaminoglycan composition contributes very little (or nothing) to urinary glycosaminoglycans. There are many articles reporting changes in urinary glycosaminoglycan excretion in urolithiasis. Please check again and replace this reference with one that is more appropriate to the subject. Citing this reference suggests that the authors are unfamiliar with "glycosaminoglycans".
Response 1: The previous references (#115, 116) was removed.
We research some references which are suitable to explain the urinary glycosaminoglycans. We described some sentences in the Section 6.1. (Limitation section), and inserted two novel references (#115, 116).
Reviewer 2 Report
Comments and Suggestions for Authors
I am not completely satisfied with the revision of this manuscript.
Although the readability of the manuscript was increased, it remains very speculative in the conclusions. As I stated previously, the description of AGE formation is well done, but no convincing evidence is provided that HSP90 molecules adhere to epithelial cells under stress conditions, that their binding with AGE is the cause of stone formation, and that the treatment with Urocalun effectively fights these mechanisms. The sentence added at lines 273-282 enforce the above-described opinion. There, the main conclusions are introduced by the verb “believe”. The beliefs should be supported by proofs, otherwise this is not science. In addition, the mention of HSP27 al line 276 is not justified. These concepts (that all the statements that HSP90 molecules adhere to epithelial cells under stress conditions, that their binding with AGE is the cause of stone formation, and that the treatment with Urocalun effectively fights these mechanisms) should at least be expressed in the “Limitations” section, whereas in that section there are now sentences that belong to the Discussion section.
Comments on the Quality of English Language
As far as the English language is concerned, the manuscript should be revised more carefully. There are several mistakes, such as forgetting the "s" in the third person singular of the present tense (in several places) and the use of incorrect terms, such as naming “squares” rather than “boxes” the graphical sections of figures.
Author Response
Response Letter to Reviewers’ Comments
Responses to Reviewer 2
Dear Reviewer 2:
Thank you for giving us the opportunity to submit a revised draft of our manuscript titled “Advanced Glycation End-Products-modified Heat Shock Protein 90 May be Associated with Urinary Stones” to the Diseases (manuscript ID: 3335984). We appreciate the time and effort you have taken to provide valuable feedback on our manuscript; your comments have enriched the manuscript and helped us produce a more balanced account of our research.
Because Editorial Office permitted the change of article type, our previous Review article has been changed to the “Hypothesis article”.
We removed and described some sentences in 2nd revised manuscript. Please read them with Yellow Highlighted.
Comments and Suggestions for Authors
I am not completely satisfied with the revision of this manuscript.
Comment 1: Although the readability of the manuscript was increased, it remains very speculative in the conclusions. As I stated previously, the description of AGE formation is well done, but no convincing evidence is provided that HSP90 molecules adhere to epithelial cells under stress conditions, that their binding with AGE is the cause of stone formation, and that the treatment with Urocalun effectively fights these mechanisms. The sentence added at lines 273-282 enforce the above-described opinion. There, the main conclusions are introduced by the verb “believe”. The beliefs should be supported by proofs, otherwise this is not science. In addition, the mention of HSP27 al line 276 is not justified. These concepts (that all the statements that HSP90 molecules adhere to epithelial cells under stress conditions, that their binding with AGE is the cause of stone formation, and that the treatment with Urocalun effectively fights these mechanisms) should at least be expressed in the “Limitations” section, whereas in that section there are now sentences that belong to the Discussion section.
Response 1: Thank you for bringing these points to our attention. We perform that the categorization of this article is changed as “Hypothesis article” because some sentences in Section 4.4. and 5.2. have not been based on the data.
In contrast, we did not suggest that (i) HSP90 molecules adhere to epithelial cells under stress condition and (ii) their binding with AGE is the cause of stone formation in our previous manuscript.
(However, we described that AGE-modification of HSP90 may affect the generation and growth of kidney stones in Section 4.4 in our 2nd revised manuscript based on your comment).
Because our sentences in the previous manuscript occur the misleading, we have re-examined our position. In our previous manuscript, we only hypothesized that HSP90 which is produced in the human proximal tubular epithelium cell line and located on the membrane of them is modified by AGEs based on the data that the metabolite intermediate/sub-products from glucose and fructose (e.g. methylglyoxal, glyceraldehyde) are able to induce the generation of AGE-modified HSP90 [Ref. 21,22].
Heng et al. introduced that HSP90 is the adhesion related protein on HK-2 cell (human proximal tubular epithelium cell line) in their report, and revealed the expression of HSP90 was upregulated by the stimulation of the nano-COM crystal which is the model of kidney stone [Ref.12].
Although Heng et al. focused the expression of HSP90 in their study, we hypothesized that AGE-modified HSP90 is able to be associated with the stimulation of kidney stone in the human proximal tubular epithelium cell under the condition of LSRDs which excess AGEs are generated in the human organs.
In contrast, the possibility that Urocalun inhibits the generation of AGE-modified HSP90 remain unclear.
We can introduce information that (i) AGE-modified HSP90 were identified in other cells and (ii) the components in Urocalun inhibited to generation of intracellular AGEs.
However, we can not introduce that (i) HSP90 in the in the human proximal tubular epithelium cell under the condition of LSRDs which excess AGEs are generated in the human organs, and (ii) Urocalun is able to inhibit the generation of AGE-modified HSP90 in the cells.
Therefore, we hypothesized those phenom in our 2nd manuscript, and re-submit it as “Hypothesis article”.
Comment 2: Comments on the Quality of English Language
As far as the English language is concerned, the manuscript should be revised more carefully. There are several mistakes, such as forgetting the "s" in the third person singular of the present tense (in several places) and the use of incorrect terms, such as naming “squares” rather than “boxes” the graphical sections of figures.
Response 2: We correct the words in the legends in Figure 4 and 5 (square → box), and correct some sentences based on your comments.
Round 3
Reviewer 2 Report
Comments and Suggestions for Authors
As the manuscript is presented now as hypothesis, it is acceptable